# Phosphorus Removal and Carbon Dioxide Capture in a Pilot Conventional Septic System Upgraded with a Sidestream Steel Slag Filter

**Dominique Claveau-Mallet [1,2,]*** , **Hatim Seltani [1,3]** and **Yves Comeau [1]**

1. Department of Civil, Geological and Mining Engineering, Polytechnique Montréal, Montréal, QC H3C 3A7, Canada; hseltani@bionest.ca (H.S.); yves.comeau@polymtl.ca (Y.C.)
2. Department of Chemical Engineering, McGill University, 3610 University Street, Montréal, QC H3A 0C5, Canada
3. Bionest, 55, 12th Street, P.O. Box 10070, Shawinigan, QC G9T 5K7, Canada
* Correspondence: dominique.claveau-mallet@polymtl.ca

**Abstract:** The objective of this work was to demonstrate the removal of the phosphorus and carbon dioxide capture potential of a conventional septic system upgraded with a sidestream steel slag filter used in recirculation mode. A pilot scale sidestream experiment was conducted with two septic tank and drainfield systems, one with and one without a sidestream slag filter. The experimental system was fed with real domestic wastewater. Recirculation ratios of 25%, 50% and 75% were tested. Limestone soils and non-calcareous soils were used as drainfield media. The tested system achieved a satisfactory compromise between phosphorus removal and pH at the effluent of the septic tank, thus eliminating the need for a neutralization step. The phosphorus removal efficiency observed in the second compartment of the septic tank was 30% in the slag filter upgraded system, compared to −3% in the control system. The slag filter reached a phosphorus retention of 105 mg/kg. The drainfield of non-calcareous soils achieved very high phosphorus removal in both control and upgraded systems. In the drainfield of limestone soil, the slag filtration reduced the groundwater phosphorus contamination load by up to 75%. The removal of chemical oxygen demand of the drainfields was not affected by the pH rise induced by the slag filter. Phosphorus removal in the septic tank with a slag filter was attributed to either sorption on newly precipitated calcium carbonate, or the precipitation of phosphate minerals, or both. Recirculation ratio design criteria were proposed based on simulations. Simulations showed that the steel slag filter partly inhibited the biological production of carbon dioxide in the septic tank. The influent alkalinity strongly influenced the recirculation ratio needed to raise the pH in the septic tank. The recirculation mode allowed clogging mitigation compared to a mainstream configuration, because an important part of chemical precipitation occurred in the septic tank. The control septic tank produced carbon dioxide, whereas the slag filter-upgraded septic tank was a carbon dioxide sink.

**Keywords:** hydroxyapatite; calcite; onsite wastewater treatment; PHREEQC; precipitation; groundwater contamination; septic tank; drainfield; reactive filter

## 1. Introduction

Conventional septic systems (e.g., a septic tank followed by a drainfield) are commonly employed in onsite and decentralized domestic wastewater treatment. The primary treatment by settling takes place in the septic tank, whereas the drainfield acts as a secondary treatment for biological carbon removal (Figure S1, Supplementary Materials).

Drainfields are built using in situ soil when conditions are favorable regarding a minimum distance above the water table, a minimum hydraulic conductivity and a minimum distance between septic systems and sensitive human or natural infrastructures (drinking water well, buildings, shores, etc.) [1,2]. The seepage from the drainfield infiltrates to the underlying soil. A drainfield is a nonpoint source of contamination, and is not subject to regular water quality monitoring. While drainfields are assumed to be efficient for carbon removal, they are neither intended nor designed for efficient nutrient removal, and plumes of dissolved phosphorus have been observed in the groundwater below several monitored drainfields [1]. Phosphorus is known to control algae growth in the majority of freshwater lakes in North America [1]. Phosphorus loads from domestic sources therefore contribute to eutrophication [3], leading to appreciable challenges in drinking water treatment [4], deterioration of ecosystems quality [5] and loss of recreational potential.

The general objective of this study is to propose a passive and simple upgrade of the phosphorus retention capacity of existing conventional septic systems. The proposed upgrade is based upon the use of steel slag filters, which are made of by-products from the steel industry [6]. Slag filters are economical, passive and efficient for phosphorus removal, which makes them appealing for decentralized treatment. Steel slag filters have been used for phosphorus removal in several pilot applications: secondary treatment of domestic wastewater [7] or dairy farm effluent [8], tertiary treatment of domestic wastewater [9], stormwater management [10] and lake remediation [11]. The main operational challenges for steel slag filters are exhaustion and clogging, which require the occasional replacement of media [9], and the need for an additional treatment step for effluent neutralization. Steel slag filters achieve a high phosphorus removal efficiency with reported total phosphorus (TP) at the effluent of wastewater treatment systems below 1 mg P/L [6,9,11,12].

The proposed upgrade consists in a barrel-shape steel slag filter in recirculation mode fed by the effluent of the second compartment of the septic tank. This was previously tested at the bench scale with a reconstituted effluent [13], where a sidestream slag filter improved the TP retention capacity of a septic tank, with an effluent pH below 9.5. This configuration is promising for existing septic tanks compared to tertiary treatment slag filters, because the need for high-pH effluent handling is avoided, and the size of the filter is reduced. This promising recirculation configuration still involves scientific knowledge gaps, such as the need for the validation of the performance of the barrel-shape filter fed with real wastewater with high alkalinity, the assessment of the drainfield integrity when it is fed with a high pH influent and the evaluation of the $CO_2$ capture of the septic tank.

Alkaline filters such as steel slag filters are a potential $CO_2$ trap due to high pH and high Ca concentration [14]. In alkaline filters, $CO_2$ sequestration depends on the filter configuration. In horizontal subsurface flows with direct contact with the atmosphere, a passive sequestration of biological or atmospheric $CO_2$ was observed [15,16]. Active $CO_2$ sequestration is possible in configurations with a filter sealed with minimum contact with the atmosphere. In such cases, $CO_2$-enriched air from an upstream biological reactor can be used for the neutralization of a slag filter effluent [17]. Few researchers measured greenhouse gas emissions or quantified the $CO_2$ capture of alkaline filters. Kasak et al. [16] measured $CO_2$, $CH_4$ and $N_2O$ emissions in horizontal subsurface flow mesocosms filled with layers of alkaline-hydrated oil shale ash and well-mineralized peat. They found that adding oil shale ash to the mesocosms significantly reduced $CO_2$ emissions compared to peat alone. Bove et al. [17] calculated that up to 75% of the $CO_2$ produced in a secondary treatment was sequestered by the neutralization of a steel slag filter effluent with $CO_2$-enriched air from the secondary treatment.

The specific objectives of this paper are:

1. To evaluate the TP removal and $CO_2$ capture of a pilot-scale conventional septic system, upgraded with a sidestream steel slag filter fed by the effluent of the second compartment of the septic tank, compared to a control conventional septic system without slag filter;
2. To evaluate the effect of media (e.g., non-calcareous or limestone sand) on TP removal in drainfields;

3. To evaluate the effect of the septic tank effluent pH on the drainfield regarding the removal of chemical oxygen demand (COD);

4. To determine precipitation mechanisms and clogging risks in septic tanks and drainfields with or without a slag filter; and

5. To develop design criteria for slag filters in recirculation based on a pH control strategy using modeling.

The TP concentration in the seepage from the drainfields was compared with the 1 mg P/L standard recommended by Quebec regulations for tertiary treatment [18]. The target pH for the effluent of the septic tank was 9.0 to avoid affecting the biological treatment taking place in the drainfield, and to meet typical pH discharge guidelines [18].

## 2. Materials and Methods

Pilot tests were conducted at the Saint-Roch-de-l'Achigan Water Resource Recovery Facility (SRDLA WRRF-) located at Saint-Roch-de-l'Achigan, Quebec, Canada, a conventional activated sludge process with a coagulant dosage for phosphorus removal during summer. The raw wastewater was mainly from domestic sources. The raw influent was pretreated by trash and grit removal and had the following characteristics: mean daily influent flowrate of 1000 m³/d, chemical oxygen demand (COD) of 350 mg/L, carbonaceous 5-day biochemical oxygen demand (CBOD$_5$) of 100 mg/L, total suspended solids (TSS) of 135 mg/L and TP of 3.3 mg P/L.

### 2.1. Septic System Pilot Tests

Pilot tests consisted in two pilot-scale septic tanks followed by drainfields with and without a sidestream slag filter (Figure 1 and Table 1, pictures of the setup shown in Figures S2 and S3). Pilot tests were conducted in a maritime container converted into a laboratory. The temperature in the container was maintained to approximately 10 °C by heating during the winter season (t = 60 to 225 d). During the fall and spring seasons, the temperature in the container fluctuated between 10 and 15 °C (t = 0 to 60 d and t = 225 to 275 d). Screened raw wastewater from the SRDLA WRRF taken upstream of the chemical dosage point was pumped into a 10 m³ septic tank (reactor R1). This settled effluent was used as the influent, and was representative of the effluent of the first compartment of a domestic septic tank. The mean influent composition is presented in Table 2.

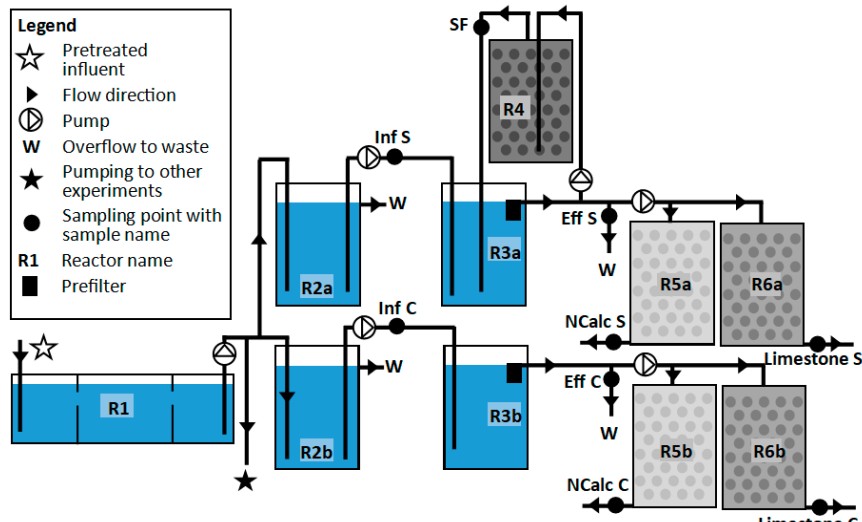

**Figure 1.** Schematic of the septic system pilot tests. Pretreatment refers to trash and grit removal. Inf C: influent control. Inf S: influent slag. SF: slag filter. Eff C: effluent control. Eff S: effluent slag. Limestone C: limestone control. Limestone S: limestone slag. NCalc C: non-calcareous control. NCalc S: non-calcareous slag.

**Table 1.** Description of reactors in the septic system pilot tests.

| Reactor | Description | Volume | Influent Flow Rate | Empty Bed Contact Time |
|---|---|---|---|---|
| | | L | L/d | d |
| R1 | Three-compartment septic tank | 10,200 | 5000 to 10,000 | 1.0 to 2.0 |
| R2a | Pumping reservoir [1] | 200 | 280 | 0.7 |
| R2b | Pumping reservoir [1] | 200 | 280 | 0.7 |
| R3a | Septic tank [2] | 200 | 180 | 1.1 |
| R3b | Septic tank [2] | 200 | 180 | 1.1 |
| R4 | Steel slag filter | 180 | 45 to 135 | 1.3 to 4.0 |
| R5a | Non-calcareous drainfield | 220 | 6.8 | 32.4 |
| R5b | Non-calcareous drainfield | 220 | 6.8 | 32.4 |
| R6a | Limestone drainfield | 220 | 6.8 | 32.4 |
| R6b | Limestone drainfield | 220 | 6.8 | 32.4 |

[1] The R2a and R2b effluent represents the effluent of the first compartment of a conventional domestic septic tank.
[2] R3a and R3b reactors represents the second compartment of a conventional domestic septic tank.

**Table 2.** Characteristics of the primary effluent for the pilot tests (mean ± standard deviation (SD) at sampling points Inf S and Inf C). Inf S: influent slag. Inf C: influent control.

| Parameter | Units | Value |
|---|---|---|
| COD | mg/L | 224 ± 73 |
| TSS | mg/L | 60 ± 40 |
| VSS | mg/L | 35 ± 19 |
| $NH_4^+$ | mg N/L | 20 |
| TP | mg P/L | 3.7 ± 1.2 |
| o-$PO_4$ | mg P/L | 2.4 ± 1 |
| $Ca^{2+}$ | mg/L | 136 ± 54 |
| $Na^+$ | mg/L | 80 |
| $K^+$ | mg/L | 9 |
| $Mg^{2+}$ | mg/L | 31 |
| $F^-$ | mg/L | 14 |
| $Cl^-$ | mg/L | 131 |
| $NO_2^-$ | mg N/L | <0.1 |
| $NO_3^-$ | mg N/L | <0.1 |
| $SO_4^{2-}$ | mg S/L | 26 |
| pH | - | 7.2 ± 0.2 |
| Alkalinity | mg $CaCO_3$/L | 425 ± 45 |

Note: parameters without a standard deviation are minor parameters that were analyzed only once.

Reactors R2 to R6 were 200 L plastic barrels. The sludge at the bottom of the pumping reservoirs (R2a and R2b) was wasted once at t = 120 d by pumping to prevent sludge accumulation. Wastewater was continuously pumped into reactors R3a and R3b, which had a prefilter at the outlet (aperture of the prefilter of 1.6 mm). Part of the R3 effluent was pumped intermittently (1 min at 40 mL per minute followed by a 7.45-min rest period) by a peristaltic pump into either a non-calcareous or a limestone drainfield. The drainfields R5 and R6 were composed of 75 cm of sand over a 12.5-cm gravel layer. The influent of R5 and R6 was pumped at a depth of 12 cm into one of four inlet tubes. Each inlet tube was used for one week sequentially to ensure a uniform division of the flow. The R5 and R6 effluents were collected in the gravel layer through four 25-mm diameter pipes with 3-mm perforations. Pilot tests were paused between days 92 and 152. During the pause, reactors R2 to R4 were kept saturated, and reactor R1 remained in operation.

One septic system had a sidestream steel slag filter fed by the effluent from the second compartment of the septic tank. The recirculation flow was set at 25%, 50% and 75% with respect to the influent flowrate, for days 0 to 100, 100 to 250 and 250 to 275, respectively. The steel slag filter was saturated with

a porosity of approximately 40% based on previous experiments with the same media [9]. The steel slag filter was fed by continuous pumping.

The systems were sampled periodically at the sampling points indicated on Figure 1 and analyzed for pH, ortho-phosphates (o-$PO_4$), TP, Ca, dissolved inorganic carbon (DIC), alkalinity, COD, TSS and volatile suspended solids (VSS), according to standard procedures [19]. Turbidity was measured instead of TSS in the drainfield effluents. Mg, K, Na and $SO_4$ were measured in the primary effluent (Table 2) by atomic absorption spectroscopy (AAnalist 200, Perkin Elmer). Cl was measured with chloride test strips (Quantab CAT 27449-40, Hach. $NH_4^+$, $F^-$, $NO_2^-$ and $NO_3^-$ were measured by ionic chromatography. $NO_2^-$, $NO_3^-$ and $NH_4^+$ were not analyzed in the test. All analyses were performed at the environment engineering laboratory of Polytechnique Montreal.

### 2.2. Slag and Sand Media

Electric arc furnace slag (3.5 mm) produced by Arcelor Mittal and provided by Minéraux Harsco (Contrecoeur, QC, Canada) was used. The slag properties were determined by Claveau-Mallet et al. [9] on a 5–10 mm sample: bulk density = 3.8 g/mL, specific surface = 0.308 $m^2$/g and chemical composition = 33% $Fe_2O_3$, 30% CaO, 16% $SiO_2$, 12% MgO, 6% $Al_2O_3$ and 3% other oxides. The slag mass in the R4 reactor was not directly measured, yet it was estimated to 410 kg based on the reactor volume (180 L) and estimated porosity (40%). The slag media was not considered a hazardous material based on toxicity characteristic leaching procedure (TCLP) tests provided by the slag supplier. The heavy metal leaching potential of this slag was assumed to be low, based on leaching tests in distilled water (Supplemental Materials, Table S1).

Two drainfields with different chemical composition (non-calcareous and limestone sands) were selected to represent two soils in Quebec, Canada, with low (non-calcareous) or high (limestone) risk of groundwater contamination by phosphorus [20]. The properties of the drainfield sands are shown in the Supplementary Materials, Table S2. Both sands were consistent with Quebec regulations for drainfields. The non-calcareous and limestone sands were obtained from the Mascouche and the Saint-Dominique quarries (Quebec, Canada), respectively. The gravel size at the bottom of drainfields was 14 to 40 mm.

### 2.3. Modeling of Septic Tank with Sidestream Slag Filters

The modeling was conducted using MATLAB and IPHREEQCom modules [21]. First, equilibrium curves of P mineral phases and the calcium carbonate saturation index were calculated to define precipitation in the septic tank. Second, the septic tank with a sidestream slag filter was simulated by mixing the septic tank influent and the steel slag filter effluent. Ca, pH and DIC in the effluent of the septic tank were simulated. Third, $CO_2$ fluxes in the septic tank were calculated based on a $CO_2$ gradient between the septic tank water and air headspace.

2.3.1. Production of Theoretical Equilibrium Curves of Phosphorus Mineral Phases

A solution was specified in the REACTION datablock using various concentrations of $CaCl_2$, NaOH, $KH_2PO_4$, $K_2HPO_4$ and $FeCl_2$. Then, hydroxyapatite or vivianite was allowed to precipitate (but not to dissolve) using the EQUILIBRIUM_PHASES datablock. The pH, Ca/Fe and o-$PO_4$ concentration after equilibration was recorded. The saturation index was computed to ensure that equilibrium was reached. The saturation index was always very low between $-10^{-15}$ and $10^{-15}$, indicating that an equilibrium was attained (a positive saturation index indicates supersaturation, while a negative saturation index indicates undersaturation, and zero indicates equilibrium [22]). The dissociation equations and solubility constants of hydroxyapatite and vivianite are provided in Table 3.

**Table 3.** Dissociation equations of phosphate mineral phases.

| Phase | Dissociation Equation | Solubility Constant |
|---|---|---|
| Hydroxyapatite | $Ca_5(PO_4)_3OH = 5Ca^{2+} + 3PO_4^{3-} + OH^-$ | $10^{-46}$ [23] |
| Vivianite | $Fe_3(PO_4)_2 : 8H_2O = 3Fe^{2+} + 2PO_4^{3-} + 8H_2O$ | $10^{-36}$ [22] |

The solubility constant of hydroxyapatite was set at $10^{-46}$ according to the fine-particle theory [3]. This value is in agreement with the equilibrium state observed at the effluent of steel slag filters [23]. The fine-particle hydroxyapatite solubility constant is eleven orders of magnitude more soluble than the tabulated bulk solubility of $10^{-57}$ [3].

### 2.3.2. Calculation of Calcium Carbonate Saturation Index

The saturation index of calcium carbonate in the influent and effluent of the septic tanks was calculated for each water sample with simultaneous pH, Ca and DIC measurements. The sample characteristics were reproduced with the REACTION datablock using various concentrations of $CaCl_2$, NaOH, HCl and $NaHCO_3$. The saturation index of calcium carbonate was calculated using PHREEQC using either crystalline calcite ($CaCO_3$, $\log(K_{sp}) = -8.48$, according to the PHREEQC database) or calcium carbonate monohydrate ($CaCO_3H_2O$, $\log(K_{sp}) = -7.144$, from the MINTEQ database [24]).

### 2.3.3. Simulation of Septic Tank with a Sidestream Steel Slag Filter

The septic tank influent, steel slag filter effluent and septic tank effluent were simulated according to the procedure presented in Table 4. Solubility constants of hydroxyapatite and calcium carbonate were set at $10^{-46}$ and $10^{-7.14}$, respectively. During each simulation, the calculated pH, o-$PO_4$, Ca, DIC and alkalinity were computed by PHREEQC. Two approaches were employed: first, a calibration was conducted with influent concentrations that represented the tests at the 50% and 75% recirculation ratios. Second, scenarios were simulated with a specified alkalinity of the septic tank influent and a specified pH of the effluent of the steel slag filter. MATLAB-PHREEQC functions are provided as Supplementary Materials.

**Table 4.** Septic tank and sidestream steel slag filter simulations methodology.

| Simulation | Methodology Using MATLAB and IPHREEQCom Modules |
|---|---|
| Influent | (1) Virtual solution simulated with the REACTION datablock using specified concentration of $CaCl_2$, $NaHCO_3$, $KH_2PO_4$ and $K_2HPO_4$<br>(2) Solution equilibrated with hydroxyapatite and calcite (saturation index of 0) using the EQUILIBRIUM_PHASES datablock |
| Slag filter effluent | (1) Influent reacted with CaO-0.4$CaCl_2$ using the REACTION datablock<br>(2) Solution equilibrated with hydroxyapatite and calcite using the EQUILIBRIUM_PHASES datablock<br>(3) Iterations performed until a target pH is reached at the end of the simulation (target pH 10.5 or 11.1, representative of pH in the effluent of slag filters). |
| Septic tank effluent | (1) 100% influent mixed with abc% slag filter effluent using the MIX datablock (abc% is the recirculation ratio)<br>(2) Solution reacted with $CO_2$(g) using the REACTION datablock to represent biological $CO_2$ production<br>(3) Solution equilibrated with hydroxyapatite (saturation index of 0) and calcite (saturation index of 0.6) |

### 2.3.4. Calculation of Carbon Dioxide Flux to Septic Tanks

The $CO_2$ flux to the septic tank was calculated for each water sample with simultaneous pH, o-$PO_4$, Ca and DIC measurements. The pH, Ca, o-$PO_4$ and DIC of the sample were reproduced in the REACTION datablock of PHREEQC using various concentrations of $KH_2PO_4$, $K_2HPO_4$, $CaCl_2$,

NaOH, HCl and NaHCO$_3$. The CO$_2$ flux, F$_{CO_2}$ (mol L$^{-1}$ d$^{-1}$), was calculated by PHREEQC according to Equation (1) [25]:

$$F_{CO_2} = k_La\left(K_{CO_2}P_{CO_2} - \{CO_2\}\right) \tag{1}$$

where $k_La$ is the off-gas CO$_2$ transfer coefficient in an anaerobic reactor, 7.2 d$^{-1}$ [25], K$_{CO2}$ is the Henry law constant for CO$_2$, 0.034 mol atm$^{-1}$ L$^{-1}$ (PHREEQC database), P$_{CO_2}$ is the partial pressure of CO$_2$ in the septic tank air space (atm) and {CO$_2$} is the CO$_2$ activity in water (mol/L). CO$_2$ flux was calculated assuming steady state conditions in the septic tank exposed to confined air (enriched in CO$_2$). The CO$_2$ partial pressure was fixed at various values within a range of 2000 to 10,000 ppm, representing small or large concentrations [17] in the confined air of the reactor.

## 3. Results and Discussion

Raw experimental data of the septic system pilot tests is provided in Supplementary Materials.

### 3.1. Phosphorus Removal Performance of the Upgraded Septic Tank and Drainfield

The P removal performance of the sidestream system is shown in Figure 2 for the septic tanks and Figure 3 for the drainfields. The control septic tank without a slag filter had an influent pH of 7.2 ± 0.2 which remained stable in the R3b compartment. The R3b compartment released o-PO$_4$ (Figure 2), probably due to sludge hydrolysis. The slag filter resulted in a favorable pH above 11, and an effluent o-PO$_4$ and TP below 0.1 and 0.3 mg P/L, respectively, during the 275 days of the tests. The water quality of the slag filter effluent was consistent with previous observations from the slag filters of the same media [17]. The cumulative P retention of the slag filter increased slowly up to 105 mg TP/kg slag (Figure 4), which is lower than the cumulative removal of 300–1700 mg TP/kg slag observed in six recent slag filter studies for domestic wastewater treatment [6]. The low P retention observed in the filter might be at least due to the short duration of the experiment; the filter had not yet reached its saturation. The mean P removal percentage was similar in all tested recirculation ratios (87%, 85% and 86% at the recirculation ratios of 25%, 50% and 75%, respectively).

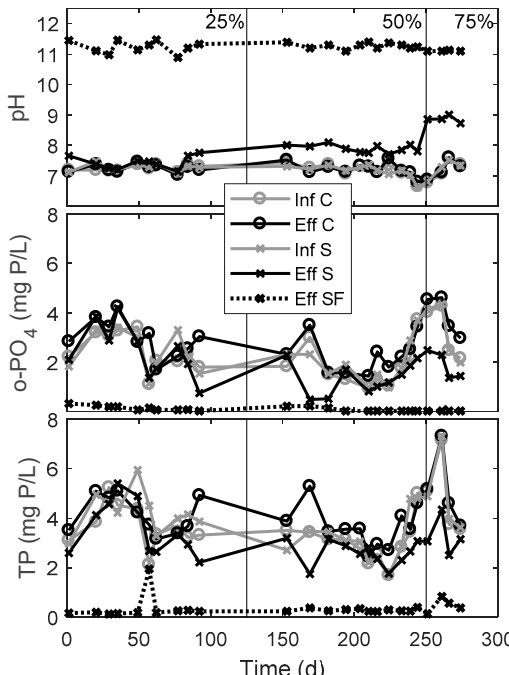

**Figure 2.** Phosphorus concentration and pH at the influent and effluent of septic tanks without (C) or with (S) a slag filter. Inf: influent, Eff: effluent; SF: slag filter. Recirculation ratios are indicated at the top of the Figure and delineated by vertical lines.

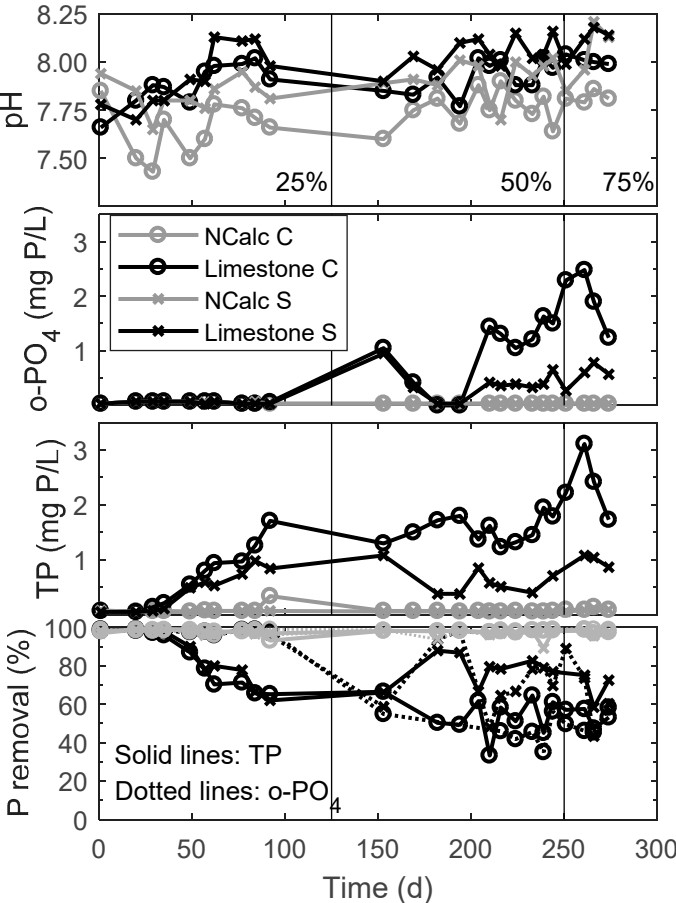

**Figure 3.** Effluent pH and phosphorus of drainfields following septic tanks without (C for control) or with a slag filter (S for slag). Recirculation ratios are indicated at the top of the Figure and delineated by vertical lines.

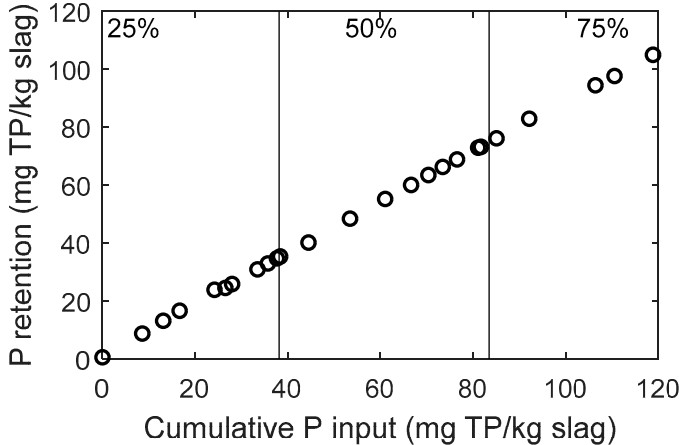

**Figure 4.** Cumulative phosphorus retention in the steel slag filter. Recirculation ratios are indicated at the top of the Figure and delineated by vertical lines.

The septic tank with a sidestream slag filter improved the TP removal efficiency compared to the control septic tank at recirculation ratios of 50% and 75% (Figure 2), reaching removal efficiencies of 11% and 32% at recirculation ratios of 50% and 75%, respectively. The o-PO$_4$ removal was markedly improved, reaching 40% at a recirculation percentage of 75% (Table 5, all data different from the control at t < 0.02, except when indicated). TP and o-PO$_4$ removal at the 25% recirculation ratio was not

significantly different from the control test. A larger pH resulted in lower TP and o-$PO_4$ concentrations in the R3a compartment of the experimental septic tank, in which the pH increased to 8.9 at the 75% recirculation ratios. The pH increase was due to the slow dissolution of the slag filter, which added hydroxide and calcium into the septic tank supernatant. Note that the P removal efficiency of the slag filter and the septic tanks was not affected by the seasonal changes in the experiment at the 25% recirculation ratio (e.g., the experiment started in the Fall, but the recirculation ratio changed to 50% in Winter). This observation contrasts with other field slag filters experiments that showed seasonal fluctuations in removal efficiency [15].

The mean TP concentration at the effluent of the septic tank with a sidestream slag filter was 3.1 mg P/L, which is not as low as other applications of alkaline filters fed with primary effluent in the main stream mode (e.g., all influent passing through the filter). Indeed, a TP concentration of 0.55 mg P/L at the effluent of a vertical flow oil shale ash filter was reported [26].

However, TP concentrations in the range of 2 to 6 mg P/L were observed at the effluent of field scale slag filters fed with the effluent of constructed wetlands [15]. In these experiments, the pH at the effluent of filters was between 8 and 9, due to a short hydraulic retention time and possibly because of the exposure to atmospheric $CO_2$. These results outline the need for a pH above 10 at the effluent of an alkaline slag filter to reach a low TP concentration.

The type of sand had an impact on the phosphorus removal in the drainfield (Figure 3). The phosphorus removal by the non-calcareous drainfield was very large in both control and steel slag filter systems, reaching a mean TP (and o-$PO_4$) concentration below 0.1 mg P/L for the 275 days of the experiment. In the limestone drainfield, however, such a high o-$PO_4$ removal was observed only for the first 100 d (o-$PO_4$ below 0.1 mg P/L at the drainfield effluent). After 100 d, the o-$PO_4$ concentration at the effluent from the limestone drainfield increased in both control and slag filter systems, possibly because of sorption saturation. Interestingly, the TP concentration in limestone drainfield effluents increased after only 25 d, even if the o-$PO_4$ concentration was still low and stable. The good removal efficiency of the non-calcareous drainfield could be explained either by its sorption capacity or by its precipitation mechanisms. The sorption capacity potential of the non-calcareous soil might not have been reached yet, and a phosphorus breakthrough could be expected after a few months or years of operation. Robertson et al. proposed, however, that equilibrium with aluminum or ferric phosphate minerals can explain the low concentrations of phosphorus monitored on a long-term [20].

**Table 5.** Mean total phosphorus removal, mean ortho-phosphate removal and pH in the second compartment of the septic tank without (control) and with a steel slag filter. Removal percentages refer to differences in concentration between the influent and effluent of the R3a and R3b reactors.

| Period | Recirculation Ratio | Mean TP Removal (%) | | Mean o-$PO_4$ Removal (%) | | pH in Septic Tank Effluent | |
|---|---|---|---|---|---|---|---|
| (d) | (%) | Control | With Slag Filter | Control | With Slag Filter | Control | With Slag Filter |
| 1 to 125 | 25 | −2 ± 10 | 17 ± 12 * | −15 ± 26 | 6 ± 23 * | 7.2 ± 0.1 | 7.4 ± 0.2 |
| 125 to 250 | 50 | −8 ± 8 | 11 ± 15 | −30 ±30 | 33 ± 26 | 7.2 ± 0.2 | 7.9 ± 0.1 |
| 250 to 275 | 75 | −7 ± 10 | 32 ± 13 | −24 ±16 | 40 ± 9 | 7.2 ± 0.3 | 8.9 ± 0.1 |

* Not statistically different from the control (t-test > 0.02, two-tailed distribution). Note: TP removal of the control system: points at t = 92 d, 224 d, 233 d and 266 d were not considered in the calculation of mean removal, as the TP concentration in the effluent was significantly greater than in the influent.

After reaching breakthrough, the TP removal efficiency of the limestone drainfield was improved by the sidestream slag filter. In the limestone control system, the mean TP removal efficiency between day 100 and day 275 was 54%, resulting in a TP concentration between 1 mg P/L and 3 mg P/L in the drainfield effluent.

In the system with a slag filter, this efficiency increased to 76% TP removal in the drainfield between day 100 and day 275, which resulted in a mean TP concentration of 0.7 mg P/L in the drainfield effluent.

A TP mass balance in the septic tank and drainfield with or without a sidestream slag filter is shown in Table 6 for the least favorable drainfield media, which was limestone sand. The calculations

were made assuming a raw wastewater influent TP concentration of 6 mg P/L, a recirculation ratio of 75% in the steel slag filter and a mature limestone drainfield (e.g., after the initial high P removal period). The system with slag filter resulted in 8% of TP in the drainfield effluent, which is significantly less than the system without the slag filter, in which 33% of the TP is released in the seepage. This represents a significant reduction of TP load to the underlying groundwater in limestone soil application: for 10,000 septic tanks, the net P capture of a slag filter upgrade is 24.3 kg P/L. According to the results of this study, a septic tank improved with a sidestream steel slag filter at a 75% recirculation ratio reaches the target of 1 mg P/L at the seepage of the drainfield, which is comparable to common nutrient removal targets in advanced secondary or tertiary treatment processes [18]. This target was not reached in the control system with the limestone drainfield, which is in agreement with previous groundwater monitoring below drainfields [20]. In drainfields located in natural, non-calcareous sand, however, efficient long-term removal of phosphorus is possible [27], and steel slag filters might not be needed in those applications.

The phosphorus recovery potential of the system was improved by the presence of the slag filter, assuming that the TP in the septic tank can be recovered in a subsequent centralized sludge treatment process [18]. In the control system, 33% of TP was accumulated in the first and second compartments of the septic tank, compared to 43% in the presence of a steel slag filter. However, the 24% TP fraction captured in the slag filter was considered unrecoverable, due to technical challenges related to phosphorus extraction from exhausted media.

**Table 6.** Comparison of total phosphorus mass balances in conventional septic systems with or without a sidestream slag filter (75% recirculation ratio; limestone drainfield). TP loads were calculated for an arbitrary reference of 10,000 individual septic systems with an influent flowrate of 1620 L/d.

| Location in System | Estimated TP Concentration (mg P/L) | |
|---|---|---|
| | Without Slag Filter | With Slag Filter |
| Influent [1] (raw domestic wastewater) | 6 | 6 |
| Effluent of septic tank first compartment [2] | 4 | 4 |
| Effluent of septic tank second compartment [2] | 4 | 2 |
| Effluent of slag filter [2] | na | 0.1 |
| Seepage of limestone drainfield [2] | 2 | 0.5 |

| Location in System | TP Load Mass Balance (kg/d and %) | |
|---|---|---|
| | Without Slag Filter | With Slag Filter |
| Influent | 97.2 (100%) | 97.2 (100%) |
| Septic tank first compartment | 32.4 (33%) | 32.4 (33%) |
| Septic tank second compartment | 0 (0%) | 9.3 (10%) |
| Slag filter | na | 23.1 (24%) |
| Drainfield (limestone) | 32.4 (33%) | 24.3 (25%) |
| Seepage reaching groundwater | 32.4 (33%) | 8.1 (8%) |

[1] Considering a removal of 2 mg/L by settling in the first compartment. [2] Considering results obtained in this study. Note: na: not applicable.

The COD and TSS removal efficiency of the drainfields with or without a slag filter was similar (Figure S4, Supplementary Materials), which indicates that the biological activity in the drainfield was not affected by the slag filter effluent. In the steel slag filter system, the R3a effluent pH did not exceed 9.0. Such pH rise is not expected to strongly inhibit $BOD_5$ removal by heterotrophic bacteria, which tolerate a pH range of 6.0 to 9.0 [18].

As the pH in the drainfield was buffered to about 8.0 by contact with atmospheric $CO_2$ and biological activity (Figure 3), biological polishing in the drainfield is not expected to be inhibited despite a septic tank effluent pH higher than 9.0. Having a slag filter had a minor impact on the effluent (seepage) pH of the drainfield (0.1 to 0.2 pH increase). The monitoring of calcium concentration, alkalinity and DIC concentration in drainfields is shown in Figure S5 as reference.

### 3.2. Inorganic Carbon Fluxes in the Septic Tank

3.2.1. Modeling Calcium Carbonate Precipitation in the Septic Tank

The pH increase in the septic tank with a slag filter may result in calcium carbonate precipitation in the septic tank. Such precipitation was observed indirectly by a reduction of DIC concentration of approximately 10 mg/L and 40 mg/L in the septic tank effluent (Figure S6, Supplementary Materials) at 50% and 75% recirculation ratios, respectively. All samples, however, were supersaturated with crystalline calcite ($CaCO_3$, $\log(K_{sp})$ = −8.48), especially those from the effluent of the septic tank with a sidestream steel slag filter (Table 7). This observation indicates that the supernatant of the septic tank with a sidestream slag filter might be in equilibrium with calcium carbonate monohydrate ($CaCO_3H_2O$, $\log(K_{sp})$ = −7.144). This compound is more soluble than $CaCO_3$, and is expected to be formed when the saturation index is close to equilibrium, as was observed in the septic tank with slag filter (Table 7).

**Table 7.** Mean supersaturation index of crystalline calcite and amorphous calcium carbonate in the conventional septic systems.

| Sampling Location | Period | Recirculation Ratio | Mean Saturation Index | |
|---|---|---|---|---|
| | (d) | (%) | Crystalline Calcite | Amorphous Calcium Carbonate |
| Influent control | 1 to 275 | na | 0.42 | −0.92 |
| Influent with slag filter | 1 to 275 | na | 0.37 | −0.96 |
| Effluent control | 1 to 275 | na | 0.52 | −0.81 |
| Effluent with slag filter | 125 to 250 | 50 | 0.95 | −0.38 |
| Effluent with slag filter | 250 to 275 | 75 | 1.94 | 0.60 |

Note: na: is not applicable.

Amorphous calcium carbonate precipitation was considered to take place in steel slag filters according to a model calibration which resulted in an intermediary solubility product of $10^{-7.5}$ [23]. In this former study, the presence of crystalline calcite was confirmed by X-ray diffraction, but no mineralogical investigations were performed to detect amorphous calcium carbonate. In future mechanistic studies, it would be useful to use mineralogical analysis techniques for both crystalline and amorphous phases such as X-ray diffraction (XRD), Fourier transform infrared spectroscopy (FTIR) or X-ray absorption near edge structure (XANES) [10]. The presence of precursor amorphous calcium carbonate, even in small quantity, might control the solubility product and affect pH estimates. This mechanism is critical in decentralized wastewater treatment, where the calcium carbonate precipitation potential is important due to highly mineralized influents (e.g., often from groundwater sources).

In the control septic tank, the DIC concentration increased (Figure S6, Supplementary Materials). This increase can be explained by the biological production of $CO_2$ through anaerobic acidogenesis, which agrees with a stable COD observed in the septic tank (Figure S7, Supplementary Materials, e.g., the effect of hydrolysis and acidogenesis on COD mass balance is very low [18]). In a septic system with steel slag filter, however, the biological production of $CO_2$ may be inhibited by a too high pH.

The proposed mechanism governing DIC concentration in slag filter-upgraded septic tanks is the precipitation of calcium carbonate monohydrate at a saturation index of 0.6 with an inhibited biological production of $CO_2$. The proposed mechanism was supported by a successful model calibration using experimental data at the 50% and 75% recirculation ratios (Figure 5). The pH, DIC and calcium concentrations in the septic tank effluent were reproduced well by simulations. The calibrated biological $CO_2$ input was 21 and 0 mg/L at 50% and 75% recirculation ratios, respectively. Note that the septic tank influent and the slag filter effluent characteristics were properly calibrated, as shown in Supplementary Materials, Table S3.

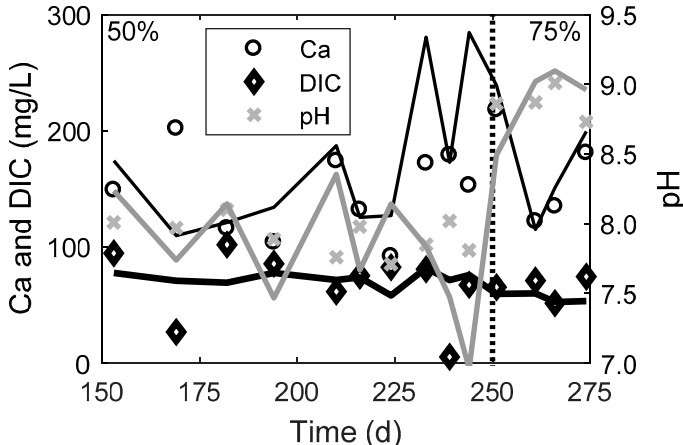

**Figure 5.** Calibration of Ca, DIC and pH at the effluent of the septic tank upgraded with a slag filter. Measurements are represented by points, and simulated data by lines. Recirculation ratios are indicated at the top and delineated by a vertical line.

### 3.2.2. Carbon Dioxide Flux into the Septic Tank

Simulated $CO_2$ fluxes in control and steel slag filter septic tanks are compared in Table 8, assuming a variety of $CO_2$ concentration in the septic tank headspace. In the control septic tank, a release of $CO_2$ was calculated at any assumed concentration in the headspace. In the septic tank with a slag filter, however, the $CO_2$ flux was significantly reduced. At 50% recirculation ratio, there was a slight release only assuming 2000 to 3000 ppm $CO_2$ in the headspace, and $CO_2$ entrapment was observed at 5000 to 10,000 ppm $CO_2$ in the headspace. At 75% recirculation ratio, the septic tank became a $CO_2$ trap at any assumed $CO_2$ concentration in the headspace. The change from $CO_2$ source to $CO_2$ sink is related to the increase of pH in the septic tank, which was 7.9 and 8.9 at 50% and 75% recirculation ratios, respectively. Assuming a $CO_2$ flux of 0.471 mol/d (Table 8), a scenario of a 2.8 $m^3$ septic tank with an influent flowrate of 1.08 $m^3$/d would result in a $CO_2$ entrapment potential of 0.27 $kg/m^3$ treated wastewater. This estimation is in the same order of magnitude of total emissions from some wastewater treatment plants [28], showing the potential of recirculated slag filters for the mitigation of wastewater treatment greenhouse gas emissions.

**Table 8.** Mean carbon dioxide fluxes in septic tanks with (50% and 75% recirculation ratios) or without slag filter (positive sign = entrapment of carbon dioxide, negative sign = release of carbon dioxide).

| Concentration in Septic Tank Headspace | $CO_2$ Flux into Septic Tank (mol/d) | | |
|---|---|---|---|
| (ppm $CO_2$) | With Slag Filter 50% Ratio | With Slag Filter 75% Ratio | Without Slag Filter |
| 2000 | −0.10 | 0.08 | −1.34 |
| 3000 | −0.05 | 0.13 | −1.29 |
| 5000 | 0.05 | 0.23 | −1.19 |
| 8000 | 0.20 | 0.37 | −1.04 |
| 10,000 | 0.29 | 0.47 | −0.95 |

Results of this study illustrated the critical role of pH in $CO_2$ capture. In future studies on slag filter-upgraded septic tanks, the analysis could be extended to methane release, an important greenhouse gas [29], as methanogenesis could be affected by the pH rise induced by the slag filter, considering an optimum anaerobic digestion pH range of 6.5 to 7.5 [30]. Methanogenesis inhibition by a high pH was achieved in microbial electrolysis cells [31]. These authors measured the methane proportion in the biogas produced in their microbial electrolysis cell. They reported a methane proportion of ~80%, ~60% and ~15% at pH 8.5, 9.5 and 10.5, respectively. As methane produced from small septic tanks is released to the atmosphere, the use of slag filters could reduce greenhouse gas emission by reducing methanogenesis in the septic tank. Partial methanogenesis inhibition in the

septic tank would result in sending more organic matter to the drainfield, where aerobic conditions for organic matter mineralization are present.

### 3.3. Phosphorus Removal Mechanisms in the Septic Tank

The o-$PO_4$-pH relationship in the septic tank effluent is shown in Figure 6. In this figure, experimental results are compared to former data of the effluent of a lab-scale septic tank with sidestream slag filter fed by the effluent of the second compartment of the septic tank [13]. Equilibrium curves of hydroxyapatite and vivianite are shown as possible P removal mechanisms by precipitation. Hydroxyapatite equilibrium curves were drawn at relevant fixed calcium concentrations: the effluent in the 2015 study had a stable calcium concentration of 30 mg/L, while most of the present study samples had a calcium concentration between 100 mg/L and 175 mg/L. Vivianite equilibrium curves were drawn at iron concentrations of 0.1 mg/L and 0.5 mg/L, which is the approximate range observed in ten different real septic tanks [20].

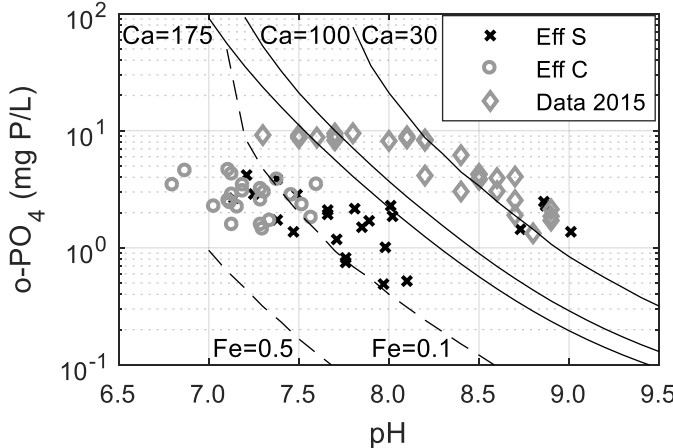

**Figure 6.** o-$PO_4$-pH relationship in septic tank effluents with sidestream slag filter fed by the effluent of the second compartment. Inf: influent, Eff: effluent; SF: slag filter. Calculated hydroxyapatite equilibrium curves are represented by full black lines (calcium concentration in mg/L next to each curve), and the calculated vivianite equilibrium curves are represented by dashed lines (iron concentration in mg/L indicated next to each curve). Data 2015: from [13] with 5–50% recirculation ratio, fed with reconstituted domestic wastewater.

The septic tank effluent o-$PO_4$ and Ca concentrations from this 2015 project were equilibrated with finely-grained hydroxyapatite for pH between 8.3 and 9.0. Results suggest that in a septic tank improved with a sidestream slag filter, phosphorus is removed by hydroxyapatite precipitation as observed in steel slag filters [17]. Note that equilibrium with finely-grained hydroxyapatite with a solubility product of $10^{-46}$ resulted in a realistic o-$PO_4$ concentrations between 0.1 mg/L and 10 mg P/L for pH between 7.5 and 9.0. The bulk hydroxyapatite solubility product (e.g., $10^{-57}$ according to [3]) is commonly considered as in a recent wastewater modeling study [32], but instead results in equilibrated o-$PO_4$ concentrations that are much lower, while supersaturation with bulk hydroxyapatite is observed in slag filters [23] or biological reactors [33].

Further mineralogical observations would be needed to confirm the presence and size of hydroxyapatite in biological reactors, and to determine an appropriate hydroxyapatite solubility product.

In this study, results were below the 100 mg Ca/L and 170 mg Ca/L finely-grained hydroxyapatite equilibrium curves, which suggests that other removal mechanisms took place. One possible mechanism is the sorption or coprecipitation of o-$PO_4$ on freshly precipitated calcium carbonate. Such a removal mechanism has been proposed by Barca et al. based on the observation of crystals of different shapes and composition at the scanning electron microscope [15]. Phosphate sorption on calcium carbonate can

have a significant effect in high-alkalinity wastewater in which significant DIC reduction is observed, which was not the case in [13] where the influent alkalinity was less than 250 mg CaCO$_3$/L, compared to 400 to 500 mg CaCO$_3$/L in this study. Calcium-carbonate-based media are known to have a phosphorus sorption capacity, such as 0.3 to 0.6 mg P/g for limestone and 3.5 mg P/g for shell sand and 0.8 mg P/g for oyster shells [34]. Assuming a recirculation ratio of 75% in the septic tank, a reduction of 20 mg/L of DIC is expected, which corresponds to 166 mg/L of calcium carbonate precipitates. Assuming a sorption capacity of 3 mg P/g, 0.5 mg P/L of phosphate is expected to sorb on calcium carbonate precipitates, which explains a part of the phosphorus removal efficiency. A second possible removal mechanism is the precipitation of iron phosphate as vivianite, which is thermodynamically possible under anaerobic conditions prevailing in a septic tank [20].

### 3.4. Effect of Influent Alkalinity on Steel-Slag-Filter Upgraded Septic Tank Operation and Costs

The recirculation ratio needed to reach a pH of 9 at the effluent of the septic tank is shown in Figure 7, based on simulations of slag filter upgraded septic tank (influent calcium concentration fixed at 175 mg/L). Two slag filter hypotheses were tested: fresh slag which is assumed at the beginning of the filter lifetime, and long-term behavior of slag according to a slow slag exhaustion. Note that simulations agree with experimental data in this study (influent of 400 to 500 mg CaCO$_3$/L alkalinity, slag filter effluent of 11.1, recirculation ratio of 75% and pH at the effluent of septic tank of 8.7 to 9.0, experimental points shown in Figure 7).

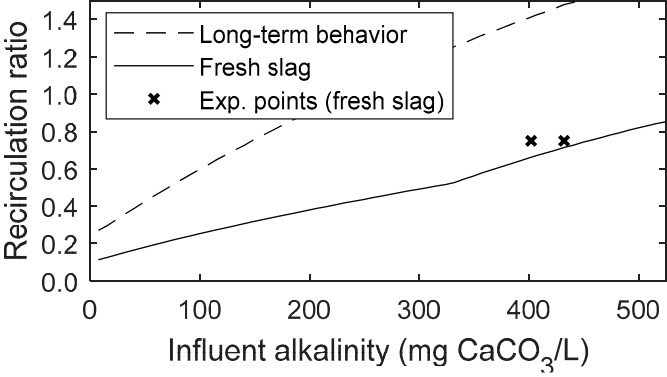

**Figure 7.** Slag filter recirculation ratio needed to reach a septic tank effluent pH of 9, with fresh slag (slag filter effluent pH of 11.1) or partly exhausted slag (pH slag filter effluent pH of 10.5). Experimental points refer to the 75% recirculation ratio phase.

The simulated needed recirculation ratio depended strongly upon the influent alkalinity and the slag freshness. With fresh slag, the needed recirculation ratio was below 70% for alkalinity up to 425 mg CaCO$_3$/L, but it increased following the slag exhaustion (needed ratio over 100% if alkalinity is above 225 mg CaCO$_3$/L). In the present study, the slag filter effluent pH decreased from approximately 11.4 to 11.1 in the 215 days of operation, and is expected to decrease progressively until slag exhaustion at a pH of approximately 10.5. Therefore, the septic tank effluent pH will decrease as well, and the phosphorus removal capacity of the septic tank will be affected. The slag filter longevity was not reached in this study, but it can be estimated using slag filter modeling.

Claveau-Mallet et al. [9] estimated the longevity of a steel slag filter operated under similar conditions (a series of two barrels of 5–10 mm slag followed by three barrels of 3–5 mm slag with a total empty bed contact time of 30 h) and fed with an influent alkalinity of 210 mg CaCO$_3$/L. The longevity was estimated at two years using simulations with the P-Hydroslag model [9]. With a higher influent alkalinity of 400 to 500 mg CaCO$_3$/L, the longevity is expected to be less than two years because of increased precipitation and clogging by calcium carbonate. The use of a sidestream slag filter instead of a flow-through slag filter, however, increases the expected longevity to approximately 18 months as

a significant part of calcium carbonate precipitation takes place in the septic tank instead of the slag filter. In this study at 75% recirculation ratio, 30% of influent DIC was removed in the septic tank, which means that the septic tank influent alkalinity was reduced by 30% compared to the septic tank influent. Such calcium carbonate control has an important impact on steel slag filter applications in onsite and decentralized treatment, where high-alkalinity influents are expected from some drinking water from groundwater supplies.

A simplified cost analysis of the implementation of septic systems for the treatment of domestic wastewater from a 3-bedrooms dwelling was conducted. Three scenarios were considered: (1) a conventional septic system without the slag filter upgrade; (2) conventional septic system upgraded with a sidestream slag filter and fed with a low-alkalinity influent (50 mg $CaCO_3$/L); and (3) conventional septic system upgraded with a sidestream slag filter and fed with a high-alkalinity influent (200 mg $CaCO_3$/L). The capital costs of the conventional septic system were spread over 20 years, which is a realistic lifetime for a septic system. The capital costs of a conventional septic system including installation were estimated to $7500 (CAD) based on local price estimations. The slag filter upgrade was scaled up to a full-scale 3-bedroom dwelling, assuming an underground concrete reactor containing ten barrels in series [9]. The analysis considered maintenance operations for the conventional septic system (sludge removal in the septic tank once every two years) and the slag filter upgrade (replacement of all barrels at fixed frequency). The barrel replacement frequency was estimated for each scenario based on the influent alkalinity and the hydraulic retention time of voids. The needed recirculation ratio was 40% in scenario 2 and 90% in scenario 3 (from Figure 7), resulting in the hydraulic retention time of voids of 40 h and 18 h, respectively. The slag filter longevity was estimated at 10 years in scenario 2, and 2 years in scenario 3, based on previous longevity predictions [9]. The resulting yearly costs and costs per removed unit of phosphorus mass are shown in Tables 9 and 10.

**Table 9.** Yearly costs (CAD) of a conventional septic system without or with a slag filter upgrade. Scenario 1: septic system only. Scenario 2: septic system with a slag filter upgrade fed with a low-alkalinity influent, operated at a recirculation ratio of 40%. Scenario 3: septic system with a slag filter upgrade fed with a high-alkalinity influent, operated at a recirculation ratio of 90%.

| Expenditure Items | Scenario 1 | Scenario 2 | Scenario 3 |
|---|---|---|---|
| Capital expenditures, conventional septic system Installation of a mainstream septic tank and a drainfield | $375 | $375 | $375 |
| Operating expenditures, conventional septic system Septic tank sludge removal | $125 | $125 | $125 |
| Capital expenditures, slag filter upgrade * | | | |
| Sidestream reactor (e.g., concrete septic tank) | - | $75 | $75 |
| ten 200-L barrels | - | $10 | $10 |
| 4 tons of 3–5 mm slag | - | $22 | $22 |
| Piping and plumbing | - | $7.50 | $7.50 |
| Pump | - | $7.50 | $7.50 |
| Operating expenditures, slag filter upgrade * | | | |
| 200-L barrels | - | $20 | $100 |
| 3–5 mm slag | - | $44 | $220 |
| Piping and plumbing | - | $15 | $75 |
| Slag disposal | - | $40 | $200 |
| Total | $500 | $741 | $1217 |

* capital and operating expenditures of the slag filter upgrade do not include costs related to commercialization (costs for installation, R&D development, certification procedures, etc.).

The yearly cost of a conventional septic system was estimated at $500. The implementation of a slag filter upgrade increased the yearly cost to $741 and $1217 in low-alkalinity and high-alkalinity influents, respectively. Such a cost increase for a single dwelling is significant, but remains realistic in comparison to the costs range of tertiary-level wastewater treatment processes in decentralized

applications. The cost of the slag filter upgrade was highly influenced by the influent alkalinity, which determined the barrel replacement frequency. The cost per removed phosphorus mass was also influenced by the influent alkalinity. In scenario 2, the removal cost was $228/kg TP, which was similar than that observed in the conventional septic system ($211/kg TP). In scenario 3, however, the removal cost was much higher, reaching $374/kg TP. Note that this economic analysis did not consider costs inherent to commercialization, such as personnel time, R&D development costs, certification process costs, etc. Therefore, the real cost of hypothetical commercialized slag filter upgrades would be higher.

**Table 10.** Cost per removed unit of phosphorus (CAD) for three studied scenarios of a 3-bedroom dwelling. Scenario 1: septic system only. Scenario 2: septic system with a slag filter upgrade fed with a low-alkalinity influent, operated at recirculation ratio of 40%. Scenario 3: septic system with a slag filter upgrade fed with a high-alkalinity influent, operated at recirculation ratio of 90%.

| Item | Units | Scenario 1 | Scenario 2 | Scenario 3 |
|---|---|---|---|---|
| input P in one year | kg | 3.55 | 3.55 | 3.55 |
| discharged P below drainfield in one year | kg | 1.18 | 0.30 | 0.30 |
| retained P in one year | kg | 2.37 | 3.25 | 3.25 |
| Cost per removed P | $/kg P | $211 | $228 | $374 |
| Marginal cost of P retention gain by the slag filter | $/kg P | - | $272 | $808 |

## 4. Conclusions and Recommendations

A sidestream slag filter was proposed as an upgrade to existing conventional septic systems. The upgraded system showed increased phosphorus retention while reducing $CO_2$ emissions from the septic tank. Implementing a sidestream slag filter increased the septic tank effluent pH at up to 9, but this pH increase did not affect the biological treatment in the downstream drainfield. The new advancements of knowledge presented in this study are (1) the demonstration of the concept of P-removing structure in a recirculation mode in septic tanks, scaled-up to a pilot study; (2) achieving a compromise between phosphorus removal, a high septic tank effluent pH and preserving the integrity of biological treatment in the drainfield; (3) the demonstration of clogging mitigation by the recirculation mode because part of the chemical precipitation occurs in the septic tank instead of the slag filter; (4) the demonstration of side benefits by $CO_2$ capture and potentially methanogenesis inhibition and (5) the demonstration that P-removing structures in recirculation mode can be designed efficiently by simulation tools.

*4.1. Recommendations for Process Design*

Implementing sidestream steel slag filters in conventional septic systems is recommended for applications where limestone sand is present. In such cases, the amount of phosphorus released to the groundwater could be reduced by as much 75%. In the presence of non-calcareous sand, slag filters may not be needed, because the sand has a phosphorus retention capacity, and a negligible concentration of soluble phosphorus is expected to be detected in the groundwater [35].

In practice, properly characterizing the aquifer and groundwater movement below the drainfield may not be possible or economical, and relying on a steel slag filter remains a safe option even in non-calcareous sand soils.

Reaching a pH value of 9.0 in the effluent of the septic tank is recommended as a compromise between efficient phosphorus removal (e.g., less than 0.1 mg P/L) and leaving enough phosphorus to allow efficient biological treatment in the drainfield. The recirculation ratio needed to reach this pH should be selected by the supplier according to the influent alkalinity (Figure 7) using modeling of the septic tank. In high-alkalinity influents (e.g., 200 to 400 mg $CaCO_3$/L), a recirculation ratio of up to 100% is needed to ensure the system efficiency during the lifetime of the filter. For a slag filter empty bed contact time of 30 h, the longevity of the slag filter is expected to be approximately 18 months. One possible operation strategy would be for the maintenance staff (e.g., visiting every

6 months) to reduce the recirculation ratio in the first year of the filter lifetime to benefit from the higher reactivity of the fresh slag. The operation of slag filters with resting periods was also previously shown to regenerate the filter [8]. The configuration of the flow path and feeding piping should be optimized to minimize clogging and short-circuiting. The proposed upward-flow configuration should be compared regarding pressure head build up to other configurations, such as a barrel with baffles and coarse slag in the inlet zone [8].

Other benefits than reduced groundwater contamination arise from the implementation of a sidestream steel slag filter. First, the phosphorus recovery potential of the system is improved by the means of the phosphorus enrichment of the septic tank sludge. Second, the septic tank becomes a $CO_2$ sink instead of being a $CO_2$ source. Third, clogging risks in the drainfield are reduced because part of the DIC is removed in the septic tank instead of being sent to the drainfield.

### 4.2. Recommendations for Further Understanding and Improved Control

Implementing a slag filter results in increased calcium carbonate sludge accumulation in the septic tank second compartment. The consequences on septic tank maintenance should be assessed, especially in high influent alkalinity applications. Clogging risks in the drainfield feeding pipes due to high pH should also be assessed. The experimental septic tank could be operated and monitored at an effluent pH ranging from 8.5 to 9.5 to improve the understanding of phosphorus removal mechanisms. Reducing the septic tank effluent pH would result in reduced recirculation ratio and increased filter longevity. The effect of a high pH (e.g., 9 to 10) in the septic tank effluent on the drainfield biological activity could be studied to evaluate the extent to which the higher pH is neutralized by atmospheric $CO_2$. Finally, the long-term stability of slag and potential leaching of metals must be assessed consistently with the practical uses of septic tanks. The suitability of used slag for other valorization applications such as road construction could be assessed to improve the lifecycle of the slag. The valorization of slag in construction might be possible considering that phosphorus retained in the slag matrix is stable, with low leaching potential [15]. In such cases, the long-term leaching potential of phosphorus must be assessed.

The $CO_2$ greenhouse gas study should be extended to $CH_4$ release in the septic tank, as its carbon dioxide equivalent for greenhouse gas effect is about 25 times higher than $CO_2$, and the $CH_4$ release could be determined experimentally. The study should also be extended to the drainfield to understand the effect of the slag filter on the fate of inorganic carbon (e.g., precipitation as calcium carbonate or $CO_2$ stripping). Finally, the impact of organic matter and biological fouling on filter longevity should be assessed. Estimates of slag filter longevity using the P-Hydroslag model could be extended to consider the presence of organic matter, as this model was designed for tertiary treatment for which organic matter was not considered [9].

**Supplementary Materials:** The following are available online at http://www.mdpi.com/2073-4441/12/1/275/s1, raw experimental data (rawdata.xlsx), PHREEQC functions (launch_septictankwithslag.m and PHREEQCfct_septictankwithslag.m), Figure S1: Schematic of a conventional septic system used in decentralized domestic wastewater treatment, Figure S2: Picture of the septic system with slag filter, Figure S3: Picture of the control septic system, Figure S4: COD and turbidity monitoring at the effluent of drainfields following septic tanks without (C) or with (S) slag filter, Figure S5: Calcium, alkalinity and dissolved inorganic carbon (DIC) in the effluent of drainfields following septic tanks without (C) or with (S) slag filter, Figure S6: Calcium, alkalinity and dissolved inorganic carbon monitoring in septic tanks without (C) or with (S) slag filter, Figure S7: COD, TSS and VSS monitoring in septic tanks without (C) or with (S) slag filter, Table S1: Heavy metals concentrations in a leaching test using a 35-g 5–10 mm slag sample in 700 mL of distilled water, shaken for 24 h, Table S2: Drainfield sand properties, Table S3: Calibration of the septic tank effluent and the slag filter effluent (mean values in the 50 and 75% recirculation ratio period).

**Author Contributions:** D.C.-M.: Conceptualization, Methodology, Software, Validation, Formal Analysis, Data Curation, Writing—Original Draft. H.S.: Methodology, Validation, Investigation, Data Curation, Visualization. Y.C.: Conceptualization, Methodology, Validation, Resources, Writing—Review & Editing, Supervision, Project Administration, Funding Acquisition. All authors have read and agreed to the published version of the manuscript.

**Funding:** This research was funded by the Natural Sciences and Engineering Research Council of Canada (grant number RDCPJ476673-14), Bionest, GHD, Arcelor Mittal Produits Longs Canada, Les Produits et Minéraux Harsco and Agro Énergie.

**Acknowledgments:** The authors thank Guy Chateauneuf from GHD for initiating this project. We also thank Cristian Neagoe, Hassan Hemouzal, Simon Amiot and Pascale Mazerolle from Polytechnique Montreal and David Marien from the Municipality of Saint-Roch-de-l'Achigan for technical assistance at the Saint-Roch-de-l'Achigan WRRF. We thank Manon Leduc, Jérôme Leroy and Denis Bouchard from Polytechnique Montreal for calcium, metal and phosphorus analyses, and Étienne Boutet from Bionest for providing the slag barrels.

**Conflicts of Interest:** The authors declare no conflict of interest. The funders had no role in the design of the study; in the collection, analyses, or interpretation of data; in the writing of the manuscript, or in the decision to publish the results.

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
