# Peer review of "Phosphorus Removal and Carbon Dioxide Capture in a Pilot Conventional Septic System Upgraded with a Sidestream Steel Slag Filter"

_water, doi:10.3390/w12010275_

Round 1

Reviewer 1 Report

Water 644171 Peer Review

General comments:

The paper presented for review is an interesting study of the possibilities of improving the functioning of conventional septic tanks. The authors tackled a topic of great utilitarian significance due to the progressive degradation of aquatic ecosystems. The work is a valuable contribution to the development of practical knowledge about wastewater management in areas without collective sewage systems. The impact of sidestream steel slag filter on the efficiency of phosphorus removal and CO2 emissions by a pilot sewage treatment plant consisting of a septic tank and drainage plot was analyzed. The results obtained indicate the great potential of the modification used to improve the functioning of this type of facilities.

The work is written clearly, concisely and exhaustively describing the research methodology and results. However, I think that the authors have described the scale of environmental problems determining their research too briefly. I encourage them to demonstrate the greater importance of research by providing a deeper insight into the problem of eutrophication. This and other small manuscript hints are provided below.

Abstract

The abstract is clearly written, stating the purpose, general methods, and results of the study.

Introduction

l. 42-43. This sentence is very simplistic. At least it should be pointed out that these are surface water ecosystems. In addition, I believe that the introduction should be supplemented with at least basic information on the risks arising from the inflow of excessive phosphorus loads into waters. You refer to one local example describing the problems of reclamation of an eutrophied lake, while the problem is global. This paragraph should be a little more worldwide description of the environmental, economic and legal consequences of phosphorus discharge to the environment, especially in the context of lakes degradation, which are much more susceptible to degradation than rivers. In this way, you will better present the validity of your research and give it a more international character.

Material and Methods

l. 111.  Providing an approximate frequency of sludge removal will facilitate the reader's practical assessment of the properties of influent.

l. 123. There is some inaccuracy between the number of days with 75% recirculation rate given here and the data in the charts. For example, in Figures 2 and 3, the vertical line runs clearly several days before the value of 250. Due to the short duration of the last phase of the study (theoretically 25 days) this may be a significant difference and it is worth checking if this is a mistake on the graphics.

l. 130-131. Table 2 shows the water parameters for testing. Some of them do not have a specific standard deviation. According to the table description, it is not known at what location and with what frequency these samples were analyzed.

l.133-139. Did you examine the waste water temperature during the experiments? If so, this information should be provided. First of all, it is important when interpreting the results of testing the efficiency of drainfields.

l. 144. There is a lack of information whether these are the dominant soil types in the area in question or only some of the commonly occurring ones. Thus, the algorithm for selecting drainage field variants has not been precisely defined.

Results and discussion

l. 226. Table 5. Was the average phosphorus removal calculated on the basis of the difference in concentrations of this element at measurement points before and after the test object? Literature often assumes% nutrient removal in a wastewater treatment plant as a reduction of the pollution load. I suppose your research is about reducing phosphorus levels (concentrations), but it is not clear to me and may not be clear to article readers.

l. 228. If possible, indicate the potential cause of this situation. It is worth analyzing this phenomenon because it has been repeated several times.

l. 233. Please check the correctness of the data in chart 3. Several times TP concentrations in the second phase of the study (50% recirculation) reach zero in the Limestone S variant. At the same time, o-PO4 are present in the effluent. Please also check the correctness of entering the phosphorus concentration data on day 233 (line 96) in the supplement's spreadsheet file.

l.262. Can you give a probable reason for the lower efficiency of limestone sand compared to silica sand in your research?

l.269-271. The sentence is unclear.

l.272-275. Was the mass balance presented in the table only based on the phosphorus concentrations obtained in the experiments, or did it also take into account the volume of flow, and therefore relates to the phosphorus charge?

l. 359. Figure 5. Although we can guess, there is no explanation of the abbreviations Eff C and Eff S in the description of the drawing.

l.384. A sorption value of 800 mg / g in oyster shells would be very desirable for wastewater management, but this is not possible.  However,  in the cited article Vohla et al. report 833.3 mg / kg shell weight. It's actually good news for natural oyster populations. Please note the mistake and correct your sentence.

Notes on the supplement:

Table S1. D60 values are missing.

Fig. S1. Please correct the name of 6a reactor.

Fig. S5. Was the sharp increase in TSS on day 233 of study (Fig. S5) associated with manipulation of sludge removal from R2a and R2b reactors? If so, this could explain the problem of estimating the phosphorus concentration at this time (heterogeneous effluent material).

Reviewer 2 Report

The manuscript reports on study on phosphorus removal and carbon dioxide capture in a pilot septic tank system upgraded with a side stream steel slag filter. The study is well planned and well conducted and the scientific tools applied are highly meaningful.

Nevertheless, certain improvements are necessary before publication is considered.

Information on the slag filter design is very scarce and not sufficient as it is the essential part of the research work. “The slag properties were determined by Claveau-Mallet et al. [9].” Provision of slag properties via a literature reference makes interpretation of the result very difficult. Principal information has to be provided in the text. Basic data e.g. mass of slag used is missing.  

The issue of disposal/subsequent use of the slug is not addressed. Relevant considerations need to be provided. Moreover, steel slag as an industrial waste material may contain significant amounts of heavy metals which can be released depending on the prevailing conditions (on spot or during later use).

Scientific literature is not commenced sufficiently. There is plenty of research and practical reports on the utilization of steel slug for P removal in wastewater treatment. The authors almost completely ignore the information provided in scientific literature. A comprehensive comparison of the gained results with published date should be included to demonstrate in how far the presented work goes beyond the current state of knowledge/science.

I disagree with the argumentation that the slug filter may serve as an efficient CO2 sink. First of all, I doubt that the CO2 amounts adsorbed are relevant regarding the typical footprint of a person. Otherwise this should be demonstrated by the authors. Secondly, I presume that the CaO in the steel slag will also adsorb CO2 in any alternative application and that there is no additional netto CO2 adsorption. I may follow the argument that the pH shift has an impact methanogenesis and may therefore reduce greenhouse gas emissions. However, this is just speculation by the authors and they do not provide scientific data that support this conjecture.
